# Role of Endothelin-1 in Right Atrial Arrhythmogenesis in Rabbits with Monocrotaline-Induced Pulmonary Arterial Hypertension

**DOI:** 10.3390/ijms231910993

**Published:** 2022-09-20

**Authors:** Yen-Yu Lu, Fong-Jhih Lin, Yao-Chang Chen, Yu-Hsun Kao, Satoshi Higa, Shih-Ann Chen, Yi-Jen Chen

**Affiliations:** 1Division of Cardiology, Sijhih Cathay General Hospital, New Taipei City 22174, Taiwan; 2School of Medicine, Fu-Jen Catholic University, New Taipei City 24257, Taiwan; 3Department of Biomedical Engineering, National Defense Medical Center, Taipei 11490, Taiwan; 4Graduate Institute of Clinical Medicine, College of Medicine, Taipei Medical University, Taipei 11042, Taiwan; 5Department of Medical Education and Research, Wan Fang Hospital, Taipei Medical University, Taipei 11696, Taiwan; 6Cardiac Electrophysiology and Pacing Laboratory, Division of Cardiovascular Medicine, Makiminato Central Hospital, Okinawa 901-2131, Japan; 7Heart Rhythm Center, Division of Cardiology, Department of Medicine, Taipei Veterans General Hospital, Taipei 11217, Taiwan; 8Cardiovascular Center, Taichung Veterans General Hospital, Taichung 40705, Taiwan; 9Department of Post-Baccalaureate Medicine, College of Medicine, National Chung Hsing University, Taichung 40227, Taiwan; 10Cardiovascular Research Center, Wan Fang Hospital, Taipei Medical University, Taipei 11696, Taiwan

**Keywords:** pulmonary arterial hypertension, atrial arrhythmogenesis, Endothelin-1

## Abstract

Atrial arrhythmias are considered prominent phenomena in pulmonary arterial hypertension (PAH) resulting from atrial electrical and structural remodeling. Endothelin (ET)-1 levels correlate with PAH severity and are associated with atrial remodeling and arrhythmia. In this study, hemodynamic measurement, western blot analysis, and histopathology were performed in the control and monocrotaline (MCT, 60 mg/kg)-induced PAH rabbits. Conventional microelectrodes were used to simultaneously record the electrical activity in the isolated sinoatrial node (SAN) and right atrium (RA) tissue preparations before and after ET-1 (10 nM) or BQ-485 (an ET-A receptor antagonist, 100 nM) perfusion. MCT-treated rabbits showed an increased relative wall thickness in the pulmonary arterioles, mean cell width, cross-sectional area of RV myocytes, and higher right ventricular systolic pressure, which were deemed to have PAH. Compared to the control, the spontaneous beating rate of SAN–RA preparations was faster in the MCT-induced PAH group, which can be slowed down by ET-1. MCT-induced PAH rabbits had a higher incidence of sinoatrial conduction blocks, and ET-1 can induce atrial premature beats or short runs of intra-atrial reentrant tachycardia. BQ 485 administration can mitigate ET-1-induced RA arrhythmogenesis in MCT-induced PAH. The RA specimens from MCT-induced PAH rabbits had a smaller connexin 43 and larger ROCK1 and phosphorylated Akt than the control, and similar PKG and Akt to the control. In conclusion, ET-1 acts as a trigger factor to interact with the arrhythmogenic substrate to initiate and maintain atrial arrhythmias in PAH. ET-1/ET-A receptor/ROCK signaling may be a target for therapeutic interventions to treat PAH-induced atrial arrhythmias.

## 1. Introduction

Endothelin (ET)-1, an autocrine and paracrine mediator, is the predominant ET isoform that is expressed in the cardiovascular system with inotropic and arrhythmogenic activity in cardiac muscles [1,2]. ET-1 expression and release are promoted by a variety of stimuli, including increased wall shear stress, stretch, and pressure overload, both in vitro and in vivo [3,4]. Plasma ET-1 levels are reported to be elevated in patients with diastolic dysfunction and elevated atrial pressures [5], promoting fibroblast proliferation and extracellular matrix production [6]. With an elevated ET-1 level in the atrium, electrical remodeling, and changes in atrial geometry, conduction heterogeneity, and size may predispose the atria to persistent arrhythmia. Burrell et al. demonstrated the occurrence of ET-1-induced arrhythmic contractions in human right atrial (RA) tissues obtained from patients undergoing cardiac surgery [7]. Previous studies have demonstrated that ET-1 has direct electrophysiological effects on pulmonary vein cardiomyocytes [8] and affects the calcium handling in atrial cardiomyocytes [9].

Pulmonary arterial hypertension (PAH) is characterized by a progressive increase in pulmonary vascular resistance leading to right ventricular (RV) failure [10,11]. The increase in pulmonary vascular resistance leads to increased afterload of the RV, resulting in upstream dilatation of the RA. It is thought that the enlargement of the RA in PAH reflects the advancement of the disease and potential progression to right heart failure as the elevated pulmonary artery and RV pressures are transmitted to the RA [12]. Chronic RA pressure overload and stretching may alter the atrial substrate by promoting fibrosis and local tissue heterogeneities, resulting in atrial conduction abnormalities. RA remodeling associated with PAH is characterized by structural and electrical changes that may pose significant risks for arrhythmias. The prevalence and incidence of atrial arrhythmias in PAH are less clear due to limited study numbers and the heterogeneous nature of the patient populations studied. The prevalence of atrial arrhythmias was 26–31%, and the 5-year incidence of atrial arrhythmias was 13.2–25.1% in PAH [13,14,15]. ET-1 has been implicated in the pathogenesis of PAH. ET-1 concentrations in tissues and plasma are increased in PAH, suggesting a pathogenic role in this disease [16]. ET-1 has been shown to exert arrhythmogenic effects through its effects on the ion currents in atrial myocytes [17]. However, little is known about the role of elevated ET-1 levels in PAH-enhanced atrial arrhythmias.

Monocrotaline (MCT)-treated animals develop significant PAH and marked RV hypertrophy via lung injury and chronic inflammation in pulmonary vascular diseases [18], suggesting that the RV dysfunction of MCT-treated animals may arise as a direct consequence of pressure overload. Rabbits develop severe PAH after a single injection of MCT [19], and this model mimics several key aspects of both primary and secondary forms of human PAH. Hiram R. et al. studied the occurrence of atrial fibrillation in PAH-induced right heart disease and concluded that RA substrate modification from atrial fibrosis plays an important role in the maintenance of atrial fibrillation [20]. However, it remains unclear whether ET-1 contributes to the pathogenesis of PAH-enhanced atrial arrhythmias. This study investigated whether MCT-induced PAH may create a proarrhythmic substrate in the RA and evaluated the role of ET-1 in the pathophysiology of PAH-related atrial arrhythmogenesis.

## 2. Results

### 2.1. Pulmonary Arteriole and RV Remodeling in MCT-Induced PAH Rabbits

In MCT-treated rabbits, there were thick-walled pulmonary arterioles with endothelial and vascular smooth muscle cell proliferation, and infiltrated macrophages within the alveolar space were observed in the MCT-treated rabbit lungs compared to the normal vascular structure in the control rabbit lungs (Figure 1A). The mean cell width and cross-sectional area of RV myocyte were greater in the MCT-treated group compared to the control group (Figure 1B). The changes in pulmonary artery pressure were assessed by measuring the RVSP via right heart catheterization. In the MCT-induced PAH group, the RVSP was significantly higher than that in the control group (Figure 1C).

### 2.2. Action Potential (AP) Morphology of the RA and Electrical Activity in Sinoatrial Node (SAN)-RA Preparations

As shown in Figure 2, RA preparations from MCT-induced PAH rabbits had a longer AP duration at repolarization of 90% (APD_90_) than those from control rabbits, and RA preparations from control and MCT-induced PAH rabbits had similar APD at repolarizations of 20% and 50% (APD_20_ and APD_50_), AP amplitude (APA), and resting membrane potential (RMP). ET-1 did not change the morphology of the AP in RA preparations from MCT-induced PAH rabbits.

Compared to SAN–RA preparations from control rabbits, SAN–RA preparations from the MCT-induced PAH group had faster beating rates, and ET-1 had more negative chronotropic actions than those from the MCT-induced PAH group (Figure 3). Among control rabbits, there were no premature beats or conduction blocks found in SAN–RA preparations treated with or without ET-1. As shown in Figure 4A, the incidence of conduction block was higher in SAN–RA preparations from the MCT-induced PAH group than those from the controls (6 of 9 vs. 0 of 7, *p* < 0.05). Only one SAN–RA preparation from the MCT-induced PAH group showed premature atrial beats with retrograde conduction from the RA to the SAN (Figure 4B), and ET-1 could induce more premature beats or short runs of atrial tachycardia (1 of 9 vs. 7 of 9, *p* < 0.05) in SAN–RA preparations from the MCT-induced PAH group. Moreover, ET-1-induced short runs of atrial tachycardia in MCT-treated SAN–RA preparations were associated with premature beat-induced reentrant tachycardia (Figure 5A,B).

### 2.3. Effects of BQ 485 on Electrical Activity on ET-1-Treated SAN–RA Preparations from MCT-Treated Rabbits

As shown in Figure 6, the administration of BQ 485 (100 nM) can mitigate ET-1-induced arrhythmogenesis of SAN–RA preparations from the MCT-induced PAH group. Among the MCT-induced PAH group with ET-1-induced triggered beats or burst firing, BQ 485 (100 nM) decreased the number of ‘triggered beat’ or ‘burst firing’ from 5 in 5 to 1 in 5 (*p* < 0.05). However, BQ 485 (100 nM) did not significantly change the conduction property among SAN–RA preparations with conduction blocks (3 of 3 vs. 1 of 3, *p* > 0.05).

### 2.4. Western Blotting Analysis

Compared to control RA preparations, MCT-induced PAH RA preparations had a reduced expression of connexin 43 but increased expressions of RhoA/Rho kinase 1 (ROCK1) and phosphorylated Akt. The expression of protein kinase G (PKG) and total Akt of RA specimens were similar between the control and MCT-induced PAH groups (Figure 7, Appendix A for expanded, uncropped gels).

## 3. Discussion

MCT-induced PAH in rabbits leads to a significant increase in RV pressure and pulmonary vascular remodeling. In the present study, the histology of lung sections from MCT-treated rabbits showed proliferated endothelial and vascular smooth muscle cells in pulmonary arterioles. Pressure overload generally results in concentric hypertrophy, characterized by an increase in cell width without affecting cell length, resulting in an increased myocyte cross-sectional area [21,22]. The MCT-induced PAH group showed an increase in the mean cell width and cross-sectional area of RV myocytes that are representative of PAH-induced RV hypertrophy. The RVSP, an indicator for PAH, was remarkably elevated in the MCT-induced PAH group compared with the control group. Accordingly, we created a reliable MCT animal model by a single injection of MCT, mimicking COPD patients who have been found to have increased pulmonary artery pressure [23]. RV pressure overload and stretching may alter the atrial substrate by promoting fibrosis and local tissue heterogeneities, which in turn predispose the patient to a risk of arrhythmia [20,24,25]. RA interstitial fibrosis increases electrical heterogeneity and steeper electrical restitution, which makes conduction block and reentry more likely [26,27,28]. Idiopathic PAH is associated with RA remodeling characterized by generalized conduction slowing with marked regional abnormalities, reduced tissue voltage, and regions of electrical silence [29]. Our present study showed that MCT created an arrhythmogenic substrate, which may promote the conduction block and trigger beats in the RA. There were several types of interactions between SAN and RA in the MCT-induced PAH group, including different types of sinoatrial block and premature atrial beats with retrograde conduction, which are predisposed to reentrant. Connexin 43 mRNA expression was significantly downregulated in pulmonary arteries in mice during hypoxia [30]. A previous study has shown that the connexin 43 mRNA in the RA was reduced by 3 weeks after MCT injection, which induces a substrate for the maintenance of atrial fibrillation due to RA reentrant activity [20].

Although PAH-induced atrial remodeling provides a substrate for the development of atrial arrhythmia, ET-1 plays a crucial role in triggering the atrial arrhythmia observed in the present study. ET-1 has been linked to cardiac electrical remodeling, which promotes changes in cardiomyocytes such as changes in ionic current density and inhomogeneous prolongation of APD, resulting in increased dispersion [31]. In this study, MCT-induced PAH rabbits had a longer APD than control rabbits. Prolongation of the APD can be responsible for the generation of the triggered activity. Prolonged APD allows L-type calcium channels to recover from inactivation, which brings an extra amount of calcium ions into the cells and triggers an AP [32]. When compared to patients in sinus rhythm, patients with chronic atrial fibrillation had a shorter APD of RA [33,34]. Therefore, the mechanism of PAH-induced atrial arrhythmias may differ from that of atrial fibrillation caused by rapid atrial pacing, which decreases the atrial effective refractory period [27]. Moreover, the left atrium serves as a substrate and is not as integral to ectopic triggers for atrial fibrillation [35]. ET-1 may develop early afterdepolarizations and oscillatory potentials that occur during the AP plateau or late repolarization in RA cardiomyocytes [36], and enhance reentrant due to the increased dispersion of APD on pro-arrhythmic substrates. Both mechanisms are involved in the genesis of ET-1-induced arrhythmias. The effects of ET-1 are mediated through interactions with two major types of cell surface receptors. ET-A receptors have been associated with electrical remodeling, vasoconstriction, and cell growth, while ET-B receptors are involved in the clearance of ET-1, inhibition of endothelial apoptosis, and release of nitric oxide and prostacyclin [31]. In this study, Western blot analysis showed an increase in ROCK1 and phosphorylated Akt expression but a similar PKG expression in RA myocytes from the MCT-induced PAH group compared to those from the control group. These results indicate that the MCT-induced PAH model did not activate the nitric oxide/cGMP/PKG signaling pathway mediated via the ET-B receptor in the RA [37]. The present study showed that the ET-A receptor antagonist BQ-485 is a beneficial drug that improves RA premature beats or burst firings in RA-SAN preparations from the MCT-induced PAH group. Therefore, blocking the ET-1-associated signaling pathway with an ET-A receptor antagonist may be beneficial for PAH-induced atrial arrhythmia. Accordingly, ET-1 plays an important role in PAH-induced atrial arrhythmogenesis, and modulation of the ET-1-related signaling pathway may exacerbate or ameliorate the occurrence of atrial arrhythmias that are correlated with severity and prognosis in patients with PAH. In both human and experimental PAH, activation of the ET-A receptor activates the phosphoinositide 3-kinases (PI3K)/Akt pathway [38,39]. Moreover, inhibition of PI3K/Akt signaling slows the beating rate of the SAN in situ and in vitro and reduces the early slope of diastolic depolarization [40]. Accordingly, ET-1 activation might increase spontaneous activity through activation of PI3K/Akt signaling and might have anti-arrhythmic effects. However, ET-1 activation decreased spontaneous activity and increased arrhythmogenesis in the MCT-induced PAH group in the present study, which means that the PIP3/Akt signaling pathway is not involved in the ET-1-induced electrical response in our MCT-induced PAH model. ET-1 also activates signals for the ROCK pathway [41], which is associated with cardiac fibrosis and arrhythmias [42,43]. ROCK inhibition improved pulmonary vascular remodeling and reduced the contraction of smooth muscle cells, thereby reducing PAH [44]. ROCK signaling activation induces connexin 43 degradation, which results in a reduction in the amount of connexin 43 [45]. As a result, PAH-induced RA arrhythmogenesis is thought to be mediated by an ET-1/ET-A receptor-mediated ROCK signaling pathway.

The data should be interpreted with caution because of the limitations of this study. First, the applied rabbit *model* of *MCT*-induced PAH may not fully *translate* to patients with PAH resulting from a *variety* of causes [46]. Second, the duration from MCT administration to performing the experiment was relatively short in our studied rabbits. Thus, late manifestations of structural or electrical remodeling in PAH may not be well demonstrated. Finally, more in vivo electrophysiological studies, qPCR, and genomic analyses are mandatory to further confirm the hypothesis of the present study.

In conclusion, the RA structure and electrical remodeling induced by a MCT-induced PAH model created an arrhythmogenic substrate with a conduction block. In PAH, ET-1 acts as a trigger factor to interact with the arrhythmogenic substrate to initiate and maintain atrial arrhythmias. The ET-1/ET-A receptor with its downstream ROCK signaling may be a target for therapeutic interventions for treating PAH-induced atrial arrhythmias.

## 4. Materials and Methods

### 4.1. Animal Model

The investigation was approved by a local ethics review board (IACUC-19-037) and conformed to the institutional Guide for the Care and Use of Laboratory Animals and the *Guide for the Care and Use of Laboratory Animals* published by the United States National Institutes of Health (NIH Publication No. 85–23, revised 1996). The MCT (Sigma, Deishofen, Germany) was dissolved in 0.5 N of HCl, and the pH was adjusted to 7.4 with 0.5 N of NaOH. Male New Zealand White rabbits (2–3 months old, 1.5~2 kg weight) received a single subcutaneous injection (60 mg/kg) of MCT or not (control group) [19].

### 4.2. Hemodynamic Analysis

At the end of 5 weeks after MCT injection, RV systolic pressure (RVSP) was recorded. The rabbits were anesthetized using an intramuscular injection of a mixture of zoletil 50 (10 mg/kg) and xylazine (5 mg/kg) and were given an overdose of inhaled isoflurane (5% oxygen; Panion & BF Biotech, Taoyuan, Taiwan) from a precision vaporizer. Adequate doses of anesthesia were confirmed with the absence of corneal reflexes and motor responses to pain stimuli, and the rabbits were placed in the supine position. A 1.4F Millar Mikro-tip catheter-introducer (Model SPR-671, Millar Instrument, Houston, TX, USA) was inserted carefully into the right jugular vein and then into the RV under pressure waveform monitoring. After a period of stabilization, RVSP was recorded using PowerLab (ADInstrument, Colorado Springs, CO, USA) [47].

### 4.3. Tissue Preparation Add Histological Analysis

After the animals were anesthetized and euthanized as prescribed previously, a midline thoracotomy was then performed and the heart and the lungs were removed after adequate doses of anesthesia were confirmed by the absence of corneal reflexes and motor responses to pain stimuli inflicted using a scalpel tip. The lungs and the RVs were immersion-fixed in neutral 10% buffered formalin and paraffin sections (5 μm) were cut. The paraffin sections were stained with hematoxylin and eosin (H&E) stain. The wall thickness and external diameter of pulmonary arterioles were measured, and relative wall thickness was calculated as [2 × wall thickness/external diameter] × 100% [22]. The cross-sectional area of RV myocytes is assessed by squaring the cell width/2 and then multiplying by 3.14. Tissue preparations of the isolated sinoatrial node (SAN)-RA (around 3 cm) were superfused with normal Tyrode’s solution consisting (in mM) of 137 NaCl, 4 KCl, 15 NaHCO_3_, 0.5 NaH_2_PO_4_, 0.5 MgCl_2_, 2.7 CaCl_2_, and 11 dextrose, with the epicardial side facing upward.

### 4.4. Electropharmacological Experiments

As described previously [48], the transmembrane Aps of the isolated RA or SAN–RA preparations were recorded within the RA and SAN–RA preparations by simultaneously using machine-pulled glass capillary microelectrodes filled with 3 M of KCl, which were connected to a WPI model FD223 electrometer under tension with 150 mg. Tissue preparations (around 3 cm) were superfused at a constant rate (3 mL/min) with normal Tyrode’s solution, consisting (in mM) of 137 NaCl, 4 KCl, 15 NaHCO_3_, 0.5 NaH_2_PO_4_, 0.5 MgCl_2_, 2.7 CaCl_2_, and 11 dextrose, saturated with a 97% O_2_/3% CO_2_ gas mixture. The temperature was maintained at 37 °C, and the preparations were allowed to equilibrate for 1 h before beginning the electrophysiological study. Electrical events were simultaneously displayed on a Gould 4072 oscilloscope and a Gould TA11 recorder. The signals were digitally recorded with a 16-bit accuracy at a rate of 125 kHz. An electrical stimulus with a 10-ms duration and suprathreshold strength (30% above the threshold) was provided by a Grass S88 stimulator through a Grass SIU5B stimulus isolation unit, and the APs were elicited by a 2-Hz electrical stimulus. The RMP was measured during the period between the last repolarization and the onset of the subsequent AP. The APA was measured from the RMP to the peak of the AP depolarization. The APD at repolarizations of 90%, 50%, and 20% of the APA were measured as the APD_90_, APD_50_, and APD_20_, respectively. ET-1 (10 nM, Sigma-Aldrich, St. Louis, MO, USA), BQ-485 (an ET-A receptor antagonist, 100 nM, Sigma-Aldrich, St. Louis, MO, USA), were superfused for 20 min to test the pharmacological responses of the RAs and the SANs. Spontaneous activity was defined as the constant occurrence of spontaneous APs in the absence of any electrical stimuli. Burst firing was defined as accelerated spontaneous activity that was faster than the basal beating activity, with the characteristics of sudden onset and termination.

### 4.5. Western Blotting Analysis

The procedure of western blotting was as described previously [49]. Tissues from the RA with or without MCT treatment were homogenized and lysed in a RIPA buffer containing 50 mM Tris, pH 7.4, 150 mM NaCl, 1% NP-40, 0.5% sodium deoxycholate, 0.1% SDS, and protease inhibitor cocktails (Sigma). The protein concentration was determined with a Bio-Rad protein assay reagent (Bio-Rad, Richmond, CA, USA). Proteins were separated in 10% SDS-PAGE under reducing conditions and electrophoretically transferred onto an equilibrated polyvinylidene difluoride membrane (Amersham Biosciences, Buckinghamshire, UK). Blots were probed with primary antibodies against connexin 43 (no. 610062; BD Biosciences, Oxford, UK), PKG (no. 3248; Cell Signaling, Beverly, MA, USA), ROCK1 (no. 4035; Cell Signaling, Beverly, MA, USA), Akt (no. 4685) (Cell Signaling, Beverly, MA, USA), phospho-Akt (no. 4060; Cell Signaling, Beverly, MA, USA), β-actin (no. ab6274; Abcam, Cambridge, MA, USA), and secondary antibodies conjugated with horseradish peroxidase. All bound antibodies were detected using an enhanced chemiluminescence detection system and analyzed with AlphaEaseFC software. All targeted bands were normalized to β-actin to confirm equal protein loading.

### 4.6. Statistical Analysis

Continuous variables are expressed as the mean ± S.E.M. A one-way repeated-measures analysis of variance (ANOVA) was used to compare the difference before and after drug administration to SAN–RA preparations and Fisher’s LSD post hoc test was used. An unpaired t-test was used to compare the differences between control and MCT-induced PAH groups. Comparisons between the different non-parametric variables were analyzed using the Chi-square test with Fisher’s exact correction. The statistical analysis was done using SigmaState 3.1, and a *p*-value of <0.05 was considered statistically significant.

## Figures and Tables

**Figure 1 ijms-23-10993-f001:**
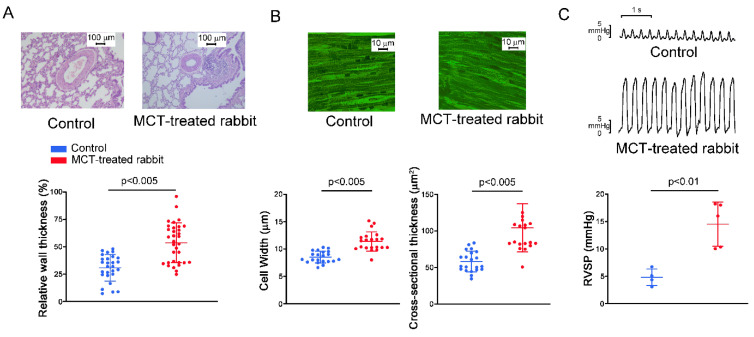
Representative pulmonary vascular, alveoli, and right ventricular (RV) histological features, and RV systolic pressure (RVSP) in the control and the monocrotaline (MCT)-treated hearts. (**A**) Hematoxylin and eosin (H&E) staining of pulmonary arterioles (original magnification × 10) and relative wall thickness of pulmonary arterioles from the control (*n* = 27 cells) and the MCT-treated (*n* = 33 cells) groups. Relative wall thickness was calculated as [2 × wall thickness/external diameter] × 100%. (**B**) Representative hematoxylin and eosin (H&E) staining of the RV (original magnification × 40) as well as cell width and cross-sectional area of RV myocytes from the control (*n* = 21 cells) and the MCT-treated (*n* = 21 cells) groups. (**C**) Representative tracings and average data of the RVSP in the control (N = 4) and the MCT-treated (N = 5) rabbits.

**Figure 2 ijms-23-10993-f002:**
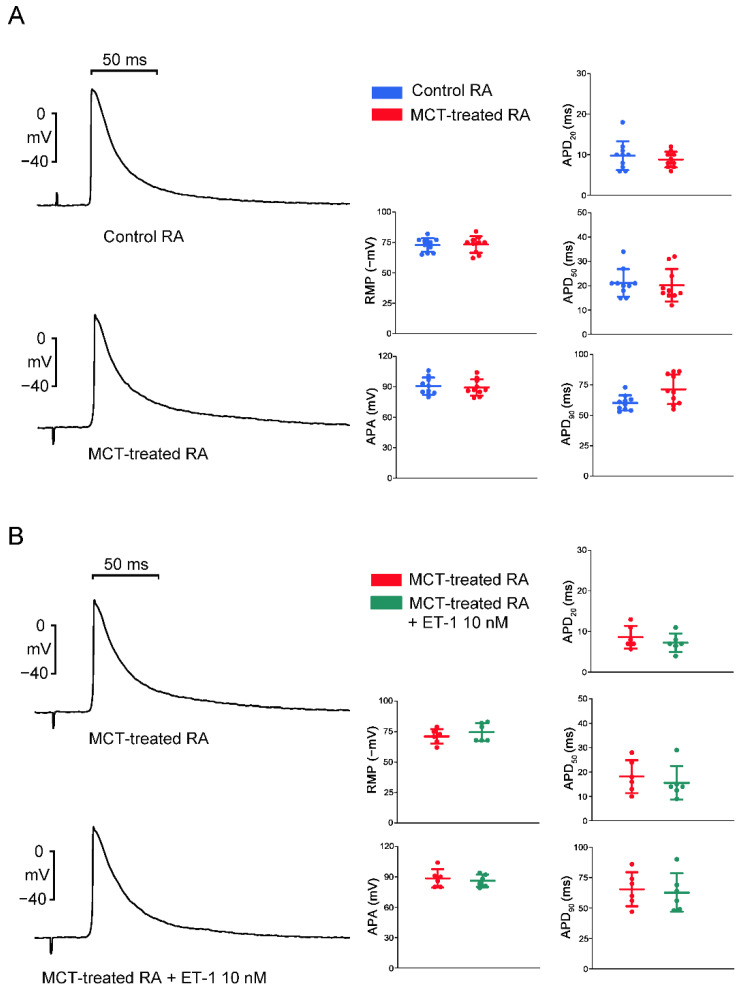
Action potential (AP) characteristics of the right atrial (RA) tissue preparations from control or monocrotaline (MCT)-treated hearts with or without endothelin (ET)-1 treatment. (**A**) Examples and average data of the APs of RA preparations from MCT-induced PAH rabbits (N = 10) and control rabbits (N = 10). (**B**) Examples and average data of the APs of RA preparations from MCT-induced PAH rabbits (N = 6) treated with or without ET-1 (10 nM). AP durations at 20%, 50%, and 90% repolarization of the AP amplitude (APD_20_, APD_50_, and APD_90_) were measured at 2 Hz; AP amplitude, APA; RMP, resting membrane potential.

**Figure 3 ijms-23-10993-f003:**
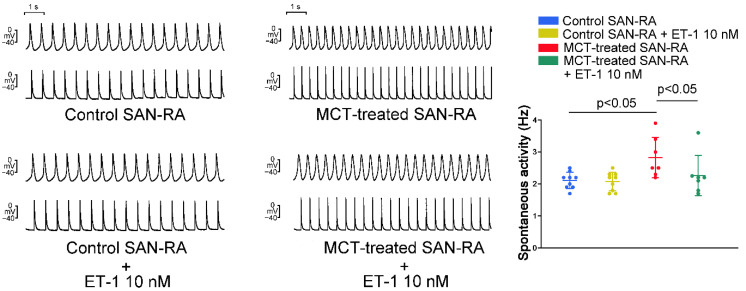
The spontaneous activity of sinoatrial node-right atrium (SAN–RA) preparations from control or monocrotaline (MCT)-treated rabbits and the effect of endothelin (ET)-1 on the spontaneous activity of sinoatrial node (SAN)–right atrium (RA) preparations from the control or monocrotaline (MCT)-treated rabbits. Representative tracings and average data of the AP morphology in SAN–RA preparations with or without ET-1 (10 nM) from the control (left panel, N = 7) or MCT-treated (right panel, N = 9) rabbits are shown.

**Figure 4 ijms-23-10993-f004:**
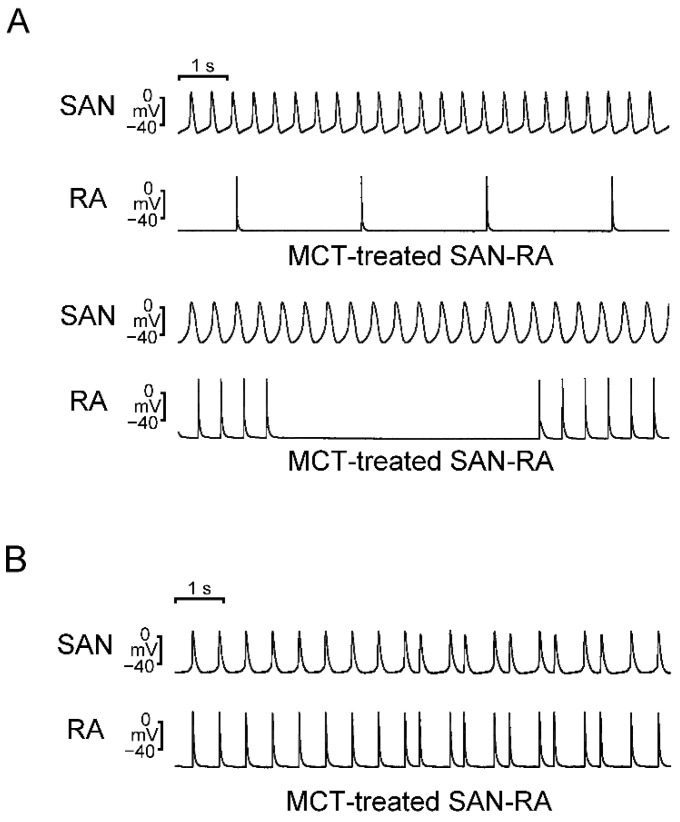
Conduction block and premature atrial beats with retrograde conduction in the sinoatrial node (SAN)–right atrium (RA) preparations from monocrotaline (MCT)-treated rabbits recorded by the conventional glass electrode. (**A**) Representative recordings of the conduction block from the SAN to the RA presented with a 6:1 conduction block (upper panel) or a long pause (lower panel). (**B**) Representative recordings of premature atrial beats with retrograde conduction from the RA to the SAN.

**Figure 5 ijms-23-10993-f005:**
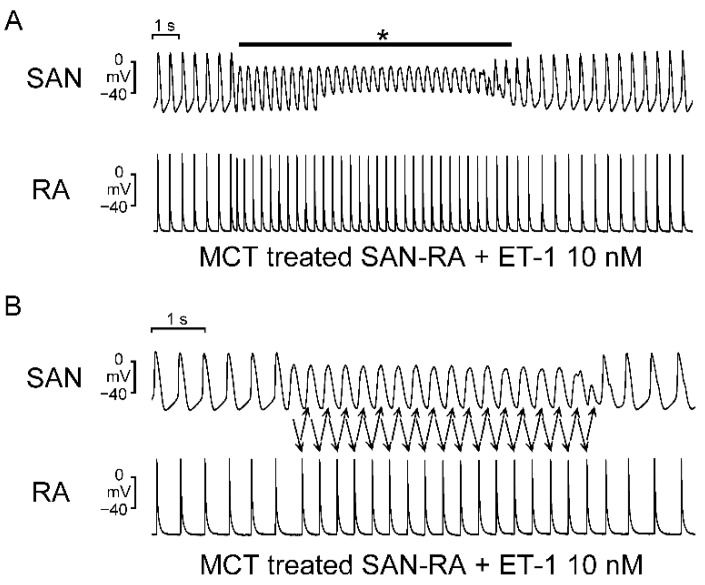
The electrical activities of endothelin (ET)-1-treated sinoatrial node (SAN)–right atrium (RA) preparations from monocrotaline (MCT)-treated rabbits were recorded by the conventional glass electrode. (**A**) A representative recording of short runs of atrial tachycardia (indicated by a black asterisk on top) in ET-1-treated SAN–RA preparations from MCT-treated rabbits. (**B**) Representative recordings revealed reentry as a mechanism of initiation of ET-1-induced short runs of atrial tachycardia in MCT-treated SAN–RA preparations.

**Figure 6 ijms-23-10993-f006:**
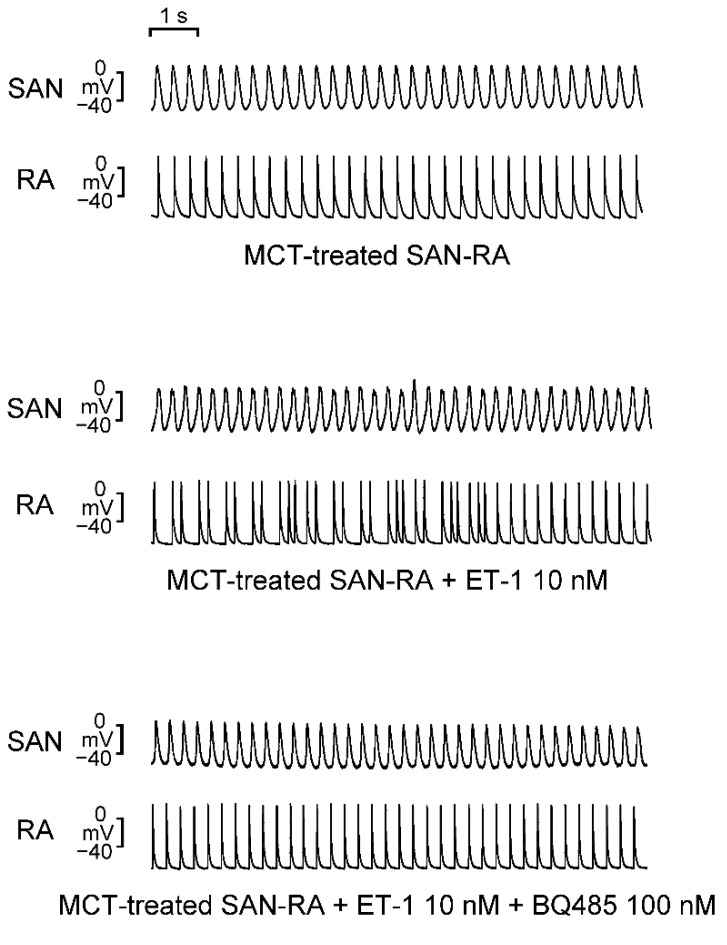
Effects of BQ 485 (an endothelin-A receptor antagonist) on the electrical activity of endothelin (ET)-1-treated sinoatrial node–right atrium (SAN–RA) preparations from monocrotaline (MCT)-treated rabbits. The representative tracings demonstrate the electrical activity of MCT-treated SAN–RA preparations after ET-1 (10 nM) and BQ 485 (100 nM) administration.

**Figure 7 ijms-23-10993-f007:**
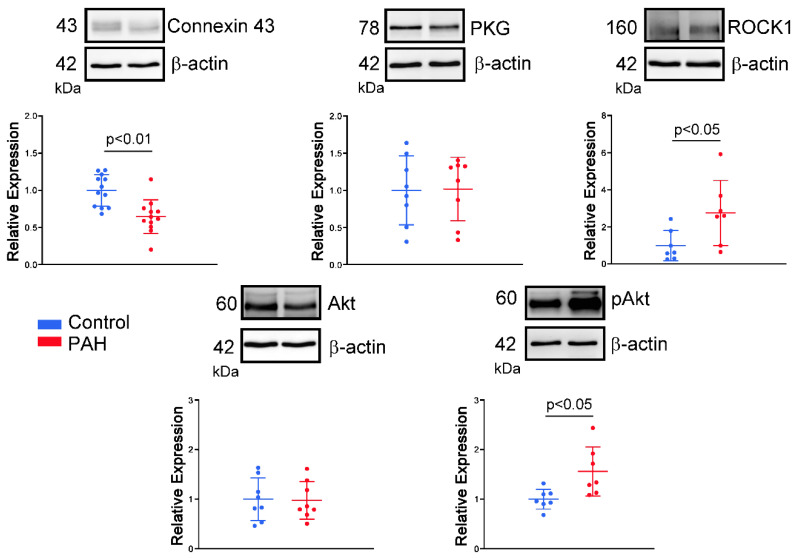
Western blot analysis of connexin 43, protein kinase G (PKG), RhoA/Rho kinase 1 (ROCK1), Akt and phosphorylated Akt (p-Akt) of right atrium (RA) tissue from control rabbits or rabbits with monocrotaline (MCT)-induced pulmonary arterial hypertension (PAH). Representative images and the relative expression of connexin 43 (N = 12), PKG (N = 8), ROCK1 (N = 7), Akt (N = 8), p-Akt (N = 7) in the control and MCT-induced PAH groups. Data are presented as the mean ± standard error of the mean.

## Data Availability

The data presented in this study are available upon request from the corresponding author.

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
