# Peer review of "Role of Endothelin-1 in Right Atrial Arrhythmogenesis in Rabbits with Monocrotaline-Induced Pulmonary Arterial Hypertension"

_ijms, 2022, doi:10.3390/ijms231910993_

Round 1

Reviewer 1 Report

The paper titled: << Role of Endothelin-1 in Right Atrial Arrhythmogenesis in Rabbits with Monocrotaline-induced Pulmonary Arterial Hyper-tension >> is authored by Lu YY et al.

The investigators studied the impact of ET-1 on the development of right atrial arrhythmias in the context of PAH induced by MCT.

The topic is of great interest. However, some concerns could be addressed before publication.

  1. Please present all bar graphs as Scatter/dot plot graphs showing mean and error bars.

  1. Although pulmonary arteries are shown, cardiac data are lacking. It would be interesting to show RV and RA hypertrophy/remodelling via histological images from RV and RA.

  1. Scale on figure 1A. is challenging to read; please increase.

  1. Why is the wall thickness expressed in Percent ? This is percent of what criteria? This parameter is usually expressed in µm. 

  1. Standardize the presentation of data by first showing control, then MCT.

  1. The information on the y axis for graphs reporting Western Blot data is curiously presented. If it is Relative to control, why control themselves have bar graphs, suggesting that they are not normalized.

Anyway, this is not the appropriate way to present western blot. Your reference should be b-actin, and control and PAC should be compared to corresponding b-actin protein bands.

7.  In figure 5, a legend is required to define what you indicate by the asterisk.

8.      Provide references in the method section when describing the animal model, hemodynamic analysis and electrophysiological study.

9.      Add a limitation section to address experiments challenges and experiments that could have been performed to improve your demonstration, including the feasibility of in vivo electrophysiological studies, or why you didn’t do qPCR or genomic analyses.

10.  Provide a highlight section with up to 5-bullet points to describe the main discoveries of the paper

11.  Provide a graphical abstract or central illustration to schematize your discovery or proposed mechanism of action leading to arrhythmia in MCT-induced atrial remodelling.

12.  You should provide full uncropped gels in the supplement. These are cropped as in the figure. This is not appropriate to verify the gel.

Author Response

Thank you very much for your detailed comments. Those comments were very instructive and very helpful to this manuscript.

  1. Regard to the general comments: “Please present all bar graphs as Scatter/dot plot graphs showing mean and error bars.”

- Thank you for your comment. According to your suggestion, we have corrected all bar graphs as Scatter/dot plot graphs showing mean and error bars in the revised figures.

  1. Regard to the general comments: Although pulmonary arteries are shown, cardiac data are lacking. It would be interesting to show RV and RA hypertrophy/remodelling via histological images from RV and RA.

-We appreciate this suggestion very much. We agree with your concern that we should describe the histological images from RV and RA to show hypertrophy/remodeling. According to your suggestion, we measured the cell width of RV myocyte and calculated the cross-sectional area of RV myocyte to investigate the impact of PAH on the RV, and PAH significantly increased the cell width and the cross-sectional area of RV myocyte. These results were presented in revised Figure 1B and described in the result (page 5, line 12-14, red font) as follows “Mean cell width and cross-sectional area of RV myocyte were greater in MCT-treated group compared to the control group.”, and we also discussed (page 8, line 2-6) as follows “Pressure overload generally results in concentric hypertrophy, characterized by an increase in cell width without affecting cell length resulting in an increased myocyte cross-sectional area (Campbell SE et al. Circ Res. 1991 and Yanucil C et al. Sci Rep. 2022). MCT-induced PAH group showed an increase of mean cell width and cross-sectional area of RV myocytes that is representative of PAH-induced RV hypertrophy.” However, we did not measure the RA hypertrophy since the effect of chronic pulmonary disease on histopathology occurs predominantly in the RV. A study done by van Oosten EM et al. demonstrated that no observable microscopic changes in the RA tissue between high and low risk obstructive sleep apnea populations (Int J Cardiol Heart Vasc. 2015).

  1. Regard to the general comments: Scale on figure 1A. is challenging to read; please increase.

-Thank you for your comment. We are sorry for the unclear presentation of the scale in figure 1A. According to your suggestion, we had corrected the revised Figure 1.

  1. Regard to the general comments: Why is the wall thickness expressed in Percent ? This is percent of what criteria? This parameter is usually expressed in µm.

-We appreciate this suggestion very much. Histopathology of vascular remodeling in PAH showed different phenotypes depending on the size and function of the different vessels, with a diameter ranging from 20-500 μm (Tobal R. et al. Front Med. 2021). Evaluation of pulmonary vascular remodeling is mostly limited to averaged increases in wall thickness, but to internal diameter decreases for vessels of different sizes, and relative wall thickness of PA was applied to evaluate pulmonary vascular remodeling in PAH (Rol N. et al. Physiol Rep. 2017). We have clarified this in the revised Figure legend as follows “Relative wall thickness was calculated as [2×wall thickness/external diameter] × 100%.”

  1. Regard to the general comments: Standardize the presentation of data by first showing control, then MCT.

-We appreciate this suggestion very much. According to your suggestion, we had changed the order of the group presentation in all revised figures.

  1. Regard to the general comments: The information on the y axis for graphs reporting Western Blot data is curiously presented. If it is Relative to control, why control themselves have bar graphs, suggesting that they are not normalized.

Anyway, this is not the appropriate way to present western blot. Your reference should be b-actin, and control and PAC should be compared to corresponding b-actin protein bands.

-Thank you for your comment. We are sorry that we did not correctly describe in the method and the presentations of Western blot in Figure 7. We have corrected the title in Y axis of the scatter plot graph to “Relative Expression”.

  1. Regard to the general comments: In figure 5, a legend is required to define what you indicate by the asterisk.

-Thank you for your comment. We are sorry that we forgot to indicate the meaning of the asterisk in the figure legend of Figure 5, and we had corrected it in the figure legend as “Representative recording of short runs of atrial tachycardia (indicated by a black asterisk on top) in ET-1-treated SAN–RA preparations from MCT-treated rabbits.” in the revised manuscript.

  1. Regard to the general comments: Provide references in the method section when describing the animal model, hemodynamic analysis and electrophysiological study.

-Thank you for your comment. According to your comment, we have provided references 19, 47, and 48 in the revised method section when describing the animal model, hemodynamic analysis, and electrophysiological study.

  1. Regard to the general comments: Add a limitation section to address experiments challenges and experiments that could have been performed to improve your demonstration, including the feasibility of in vivo electrophysiological studies, or why you didn’t do qPCR or genomic analyses.

- We appreciate this suggestion very much. According to your suggestion, we have added a limitation section to address experiment challenges and experiments that could have been performed to improve our demonstration in the revised manuscript (page 11, line 18-19 and page 12, line 1-5, red font) as follows: “The data should be interpreted with caution because of the limitations of this study. First, the applied rabbit model of MCT-induced PAH may not fully translate to the patients with PAH resulting from a variety of causes (Gomez-Arroyo JG. Et al. Am J Physiol Lung Cell Mol Physiol. 2012). Second, the duration from MCT administration to performing the experiment is relatively short in our studied rabbits. Thus, late manifestations of structural or electrical remodeling in PAH may not be well demonstrated. Finally, more in vivo electrophysiological studies, qPCR, and genomic analyses are mandatory to further confirm the hypothesis of the present study.”

  1. Regard to the general comments: Provide a highlight section with up to 5-bullet points to describe the main discoveries of the paper.

- We appreciate this suggestion very much. According to your suggestion, we provide a highlight section with up to 5-bullet points to describe the main discoveries of the paper attached in the supplement.

  1. Regard to the general comments: Provide a graphical abstract or central illustration to schematize your discovery or proposed mechanism of action leading to arrhythmia in MCT-induced atrial remodelling.

- We appreciate this suggestion very much. According to your suggestion, we provided a graphical abstract to schematize your discovery or proposed mechanism of action leading to arrhythmia in MCT-induced atrial remodeling attached in the supplement.

  1. Regard to the general comments: You should provide full uncropped gels in the supplement. These are cropped as in the figure. This is not appropriate to verify the gel.

- We appreciate this suggestion very much. According to your suggestion, we provide full uncropped gels in the supplement.

The above descriptions are the responses to your comments and suggestions.

Sincerely yours,

Yi-Jen Chen, MD, PhD

Reviewer 2 Report

The authors described a well-conducted experiment with scientific rigor. The reading is pleasant and the comments on the results are reasonable. The bibliography is coherent and sufficiently updated.

Author Response

Thank you very much for your detailed comments. Those comments were very instructive and very helpful to this manuscript.

Regard to the general comments: “The authors described a well-conducted experiment with scientific rigor. The reading is pleasant and the comments on the results are reasonable. The bibliography is coherent and sufficiently updated.”

-We are very much thankful to you for your deep and thorough review, and are grateful for your positive and encouraging comments.

Sincerely yours,

Yi-Jen Chen, MD, PhD

Reviewer 3 Report

In the paper "Role of Endothelin-1 in Right Atrial Arrhythmogenesis in Rabbits with Monocrotaline-induced Pulmonary Arterial Hypertension," the authors investigate investigated whether monocrotaline-induced pulmonary arterial hypertension (PAH) may create a proarrhythmic substrate in the right atrium and evaluate the role of endothelin-1 in the pathophysiology of PAH-related atrial arrhythmogenesis. 

The research design is appropriate, and the paper I well written

Major points

Please underscore the clinical relevance of the finding of the study

Please add limitations of the study

Minor points

Revised the manuscript for typos/error

Author Response

Thank you very much for your detailed comments. Those comments were very instructive and very helpful to this manuscript.

  1. Regard to the general comments: “Please underscore the clinical relevance of the finding of the study”

-We appreciate your suggestion very much. According to your suggestion, we underscore the clinical relevance of the finding of the study in the discussion of the revised manuscript (page 10, line 18-19 and page 11, line 1-2, red font) as follows: “Accordingly, ET-1 plays an important role in PAH-induced atrial arrhythmogenesis, and modulation of ET-1-related signaling pathway may exacerbate or ameliorate the occurrence of atrial arrhythmias that correlated with the severity and prognosis in patients with PAH.”

  1. Regard to the general comments: Please add limitations of the study:

-We appreciate this suggestion very much. According to your suggestion, we have added a limitation section to address experiment challenges and experiments that could have been performed to improve our demonstration in the revised manuscript  (page 11, line 17-19 and page 12, line 1-5, red font) as follows “The data should be interpreted with caution because of the limitations of this study. First, the applied rabbit model of MCT-induced PAH may not fully translate to the patients with PAH resulting from a variety of causes (Gomez-Arroyo JG. Et al. Am J Physiol Lung Cell Mol Physiol. 2012). Second, the duration from MCT administration to performing the experiment is relatively short in our studied rabbits. Thus, late manifestations of structural or electrical remodeling in PAH may not be well demonstrated. Finally, more in vivo electrophysiological studies, qPCR, and genomic analyses are mandatory to further confirm the hypothesis of the present study.”

  1. Regard to the general comments: “Revised the manuscript for typos/error”

-Thank you for your comment. We have checked our manuscript for typos/error before submitting the revised manuscript.

The above descriptions are the responses to your comments and suggestions.

Sincerely yours,

Yi-Jen Chen, MD, PhD

Round 2

Reviewer 3 Report

No other comments

Author Response

Responses to Reviewer #3

Thank you very much for your detailed comments. Those comments were very instructive and very helpful to this manuscript.

Regard to the general comments: “English language and style are fine/minor spell check required”

-We are very much thankful to you for your deep and thorough review and are grateful for your positive and encouraging comments. According to your suggestion, we have checked our revised manuscript's spelling and style, and the changes we made are shown in red font.
